# Reactive Black-5, Congo Red and Methyl Orange: Chemical Degradation of Azo-Dyes by Agrobacterium

Jaspreet Kaur [1,†], Gaurav Mudgal [1,*,†], Arvind Negi [2,*], Jeewan Tamang [3], Shambhawi Singh [1], Gajendra Bahadur Singh [1], Jagadeesh Chandra Bose K. [1], Sandip Debnath [4], Mohammad Ahmad Wadaan [5], Muhammad Farooq Khan [5], Janne Ruokolainen [6] and Kavindra Kumar Kesari [2,6]

1 University Institute of Biotechnology, Chandigarh University, Mohali 140413, India
2 Department of Bioproduct and Biosystems, Aalto University, 02150 Espoo, Finland
3 University Institute of Agricultural Sciences, Chandigarh University, Mohali 140413, India
4 Department of Genetics and Plant Breeding, Institute of Agriculture, Visva-Bharati University, Sriniketan 731235, India
5 Department of Zoology, College of Science, King Saud University, Riyadh 11362, Saudi Arabia
6 Department of Applied Physics, School of Science, Aalto University, 02150 Espoo, Finland
* Correspondence: gauravmdgl@gmail.com or gaurav.mudgal@cumail.in (G.M.); arvind.negi@helsinki.fi or arvind.negi@aalto.fi (A.N.); Tel.: +91-9557519824 (G.M.)
† These authors contributed equally to this work.

**Abstract:** The commercial processing of various biomaterials extensively uses azo dyes (including reactive, direct, acidic, and basic dyes). These industrial applications produce wastewater containing a large volume of solubilized azo dye and hydrolyzed by-products. The treatment of such wastewater is primarily carried out by chemical and, to an extent, physical methods, which lack selectivity and efficiency. Notably, the chemical methods employ free radicals and oxidizing agents that further increase the chemical waste and produce non-biodegradable side-products. Therefore, there is an increasing trend of using microbial-assisted methods. The current study identified a specific *Agrobacterium* strain (JAS1) that degraded the three structurally distinct azo dyes (Reactive Black 5, Methyl Orange, Congo Red). JAS1 can tolerate high concentrations and be used to perform the in-solution degradation of azo dyes, respectively: Methyl Orange (5.5 g/L and 5.0 g/L), Congo Red (0.50 g/L and 0.40 g/L), and Reactive Black 5 (0.45 g/L and 0.40 g/L). Our study elucidated the molecular mechanisms (primarily enzymatic degradation and adsorption) responsible for the JAS-1-assisted decoloration of azo dyes. The JAS-1-assisted degraded products from these azo dyes were found biodegradable as the germination and seedling growth of wheat seeds were observed. To enhance the scope of the study, JAS1-assisted decolorization was studied for cellulosic materials, indicating a potential application in de-inking and de-dyeing process in recycling industries.

**Keywords:** azo dyes; functionalized cellulose; *Agrobacterium*; decolorization; degradation

## 1. Introduction

The degradation of azo dyes produces non-biodegrade chemicals with a high environmental footprint. Their source accessibility and facile usage as chemical auxiliaries, as well as the coloring properties of azo dyes, make them suitable commercial dyes in the development of various application-oriented biomaterials [1,2]. However, their growing popularity in commercial industries and intrinsic flaws in their molecular structure (prone to easy hydrolysis) make them a challenging hazardous chemical. Typically, the hydrolysis of azo dyes produces aromatic amines and related aromatic compounds, clinically reported as mutagenic to multicellular organisms [3]. Therefore, commercial industries that are continuously using higher quantities of azo-dyes are primarily natural- or synthetic cellulose-material-based industries and often release enormous dye waste during

their processing. In most cases, the wastewater contains dissolved and hydrolyzed non-biodegradable products of azo dyes. This chemical waste has been reported to severely hamper the marine life form and a few reports even indicate the direct impact of these chemicals on human health.

Most commonly used azo dyes are direct, basic, and reactive dyes. However, these dyes typically contain sulfate groups, attributed to their high water solubility; therefore, most of the azo dye (more than 50%) remains in dissolved form and is ultimately released into the wastewater. Therefore, wastewater management plays a critical role in reducing the effects of these dyes before they are passed onto freshwater bodies. Current methods mainly depend on using additional chemicals to hydrolyze azo dyes and absorbing materials to absorb smaller by-products. However, such methods require high chemical usage; therefore, there is a growing trend of using biological agents to remove such azo dyes from wastewater.

Microbial decolorization studies with azo dyes have investigated the use of co-substrates such as glucose, yeast extract, and others, which may enhance the dye degradation rate. In addition, influence from factors such as pH, incubation temperature, and dye concentration has been routinely highlighted for optimizations [4–7]. Bacteria rarely use dyes as the sole carbon source for their growth [8]. The qualifying attributes of a dye bioremediating microbial agent are: (i) the ability to tolerate high effluent dye concentrations and (ii) the ability to convert these into stable, non-toxic by-products [9,10]. Generally, such dye-remediated solutions are put to in vitro and ex vitro testing of their toxicity profiles [11–18]. Microbial endophytes reside within plant tissue systems, many of which inherit properties of their host(s), such as the plant-growth-promoting attributes. These endophytes collectively offer nutrient-rich soil, bioavailable assimilates, and protection from various biotic and abiotic stresses [19,20].

We previously isolated and variously characterized a plant-growth-promoting (PGP) endophytic bacteria from leaf-initiated plant tissue cultures of *Sansevieria trifasciata* [21]. Besides the many PGP attributes, JAS1 allows for plant growth under intermittent soil drying by improving soil water retention, possibly due to the effect of its exopolysaccharide secretion [21]. In the present study, we tested the ability of JAS1 to decolorize and tolerate various in-solution levels of the three structurally distinct, environmentally unfriendly azo dyes, MO, CR, and RB-5. We evaluated the effect of physicochemical factors on the dye decolorization of JAS1 as well as the toxicity status of dye degradation products for plant growth. Further, we tested the applicability of seed priming with JAS1, which enables plants to cope with the detriments of dyes. We could also reproduce the dye decolorization effect of JAS1 over-dyed cellulosic substrates and documented a cellulose-caging-assisted sustainable treatment option.

## 2. Materials and Methods

All chemicals (dyes: Methyl Orange, Congo Red, Reactive Black-5; cell culture chemical) were purchased from Sigma-Aldrich, India (New Delhi), and Himedia, India (Mumbai), are of analytical grade, and were used without any further purification.

### 2.1. Bacterial Isolate and Culture Medium

*Agrobacterium pusense*, strain JAS1, was isolated from leaf-explant-initiated in vitro tissue cultures of snake plant, *Sansevieria trifasciata var. Laurentii* [Prain.] [21,22]. Its culturing over nutrient agar was supplemented with 20 mg/L ampicillin (48 h at 28 °C in the dark). A single colony was then raised over minimal salt medium (MSM; composed of 0.0902 M $Na_2HPO_4$, 0.022 M $KH_2PO_4$, 0.0187 M $NH_4Cl$, 0.0086 M NaCl, 0.0005 M $MgSO_4$, 0.0001 M $CaCl_2$ and 0.6% glucose, and pH 7.0) [23] under antibiotic selection (in the dark at 28 °C, 150 rpm). This liquid medium (MSM) and/or supplemented with 1.5% agar (MSMA) was used further in dye decolorization assays.

### 2.2. Dyes and Decolorization Assay

In our study, we considered azo dyes (Methyl Orange (MO), Congo Red (CR), and Reactive Black 5 (RB-5). In a standard decolorization test, 0.10 g/L of each dye was mixed in separate tests with 25 mL MSM (containing 20 mg/L ampicillin) and inoculated with 2% of its volume with an overnight grown starter culture raised from a single colony of JAS1. The test mixtures were incubated on an orbital shaker (at 28 °C, 150 rpm, in the dark conditions). The control tests only contained respective dyes in MSM, where inoculation with JAS1 was avoided. At regular intervals, test mixtures were visually observed for color changes. Supernatants (6000 rpm, 4 °C, 5 min) were processed for optical density values (using a Shimadzu UV-1800 spectrophotometer) at wavelengths recommended for various dyes ($OD_{464\,nm}$ for MO, $OD_{498\,nm}$ for CR, and $OD_{598\,nm}$ for RB-5). The dye decolorization percentage (DD%) was calculated as per the formula below:

$$DD\% = \frac{\text{Initial absorbance } - \text{ Final absorbance}}{\text{Initial absorbance}} \times 100$$

Dye decolorization by JAS1 was also assayed under the effects of changes in various physicochemical parameters, such as different types of carbon and nitrogen sources, variations in pH, temperature, initial dye concentration, inoculum size, and the speed (rpm) of shaking incubations. All experiments were conducted thrice, with each trial involving three replicates.

### 2.3. FTIR Analyses

To elucidate whether dye decolorization by JAS1 was specifically due to any structural changes in the dye chemical profile, the centrifugally (6000 rpm, 4 °C, 5 min) clarified culture supernatants in the above assays (collected after the 5th day, when all dye characteristic coloration in-solution was visually undetectable) were freeze-dried in a lyophilizer (make model). The powdered preparation was mixed with KBr and followed with chemical composition analysis by Fourier Transformed InfraRed (FTIR) spectroscopy (on Perkin Elmer Spectrum Two UATR) at the Department of Chemistry, Chandigarh University, following a standard approach [24].

### 2.4. Thin Layer Chromatography (TLC)

A milliliter aliquot of supernatant from dye decolorization treatments with the bacterial isolate and untreated control tests (each with 100 mg/L of dye) were collected after brief centrifugation (at 10,000 rpm, 10 min), alkalized to pH 8.0, extracted in chloroform and evaporated (45 °C, 10 min) to avail a concentrate, which was then dissolved in 1 mL chloroform. A 5 µL spot was loaded on a TLC silica gel $60F_{254}$ plate (Merck, Germany), used as a stationary phase while the mobile phase was composed of chloroform: DMSO: Isopropanol (3:3:4). Lanes were observed under both visual and UV light [25].

### 2.5. Assaying Dye Tolerance and Growth of JAS1 Exclusively on Dyes as Sole C-Source

We wished to ascertain the levels of the three azo dyes in solution, which could influence either the growth and/or decolorization efficiency of JAS1. In other words, we wished to elucidate the concentration level for each of the dyes in solution, which could be tolerated by JAS1 and above which growth in suspension could be halted, and, if so, whether this halt is reversible. In separate tests for each of the dyes, liquid MSM carrying a range of dye concentrations (100–6000 mg/L) and antibiotics was inoculated with $1 \times 10^6$ CFUs/mL of JAS1 from freshly grown overnight culture and incubated as mentioned before. The growth of JAS1 was assayed spectrophotometrically ($OD_{600\,nm}$) and dye decolorization (DD%) was recorded at suitable time intervals for two week-long treatments. The viability of JAS1 in all cases in each trial was routinely checked by subculturing a loopful inoculum from each test over NA. the growth of JAS1 solely in the presence of dyes (200 mg/L) as a carbon source was checked by simply removing glucose from MSM and spectrophotometrically elucidating the growth.

### 2.6. Caging of JAS1 and Its Cell-Free Spent Supernatant

A filter sterile spent supernatant of JAS1 culture (raised in MSM for 4 days) was subjected to dialysis against a minimal volume of sterile distilled water (SDW; pH 7.0 = permeate), enough to submerge the dialysis membrane (MWCO 12–14 kDa; Himedia, Mumbai, India). After 4 days of dialysis running under 50 rpm stirring, the retentate was mixed with 50 mg/L of dyes. For control tests, an equivalent volume of SDW (pH 7.0) was also treated with dye. The solutions were incubated at room temperature and dye decolorization was analyzed spectrophotometrically (as mentioned before for each dye) at various intervals over a week. In another experiment, 2–4 times the concentrated supernatant (after dialysis) was also analyzed for any effect on the dye decolorization rate for each dye. In yet another experiment, the above materials and methods were similarly exploited to use both JAS1 culture and/or its cell-free supernatant when testing a dye degrading system.

### 2.7. Effect of Dyes and Their Degradation Products on In Vitro Seed Germination and Growth

Doses respective to each of the three azo dye(s) that critically affect seed germination and the seedling growth profile of wheat were elucidated following a week's treatment. To do this, surface-sterilized seeds of a commercially available wheat (variety UNNAT PBW 343 purchased locally from Grain market, Mohali, Punjab, India) were soaked in aqueous solutions contained in separate p100 petri-plates with the following range of concentrations (0–1000 mg/L) respective to each of the three dyes: MO, CR, and RB-5. Seed germination percentage, shoot, and root length characteristics in the respective dye and JAS1-degraded dye solutions were recorded. The above effects were also compared to those in seeds treated with a similar volume of cell-free supernatant prepared from overnight-grown JAS1 in MSM. From these results, we inferred a particular dose ($LD_{50}$ and $LD_{100}$) respective to each of the three dyes that accounts for nearly 50% and/or 100% loss of seed germination. In another experiment, centrifugally recovered cell-free supernatants (ultrafiltered with 0.22 μm disc filters, Millipore), followed by JAS1-assisted in-solution dye degradation at various initial dye concentrations (stipulated as $LD_{100}$ in the above method) of the three dyes MO, CR, and RB-5, were used directly as the germination media on p100 Petri-plates to account for the toxicity of the degraded products over wheat seed germination and seedling growth in a week.

### 2.8. In Vitro Effect of Seeds Priming with JAS1

Wheat seeds were surface-sterilized and soaked overnight in ultra-filtered tap water (TW). A set of 15 wheat seeds were primed by co-cultivation with 20 mL of TW resuspended JAS1 culture ($1 \times 10^8$ CFU mL$^{-1}$) overnight at room temperature. The control set saw treatment with TW. After briefly washing seeds ($3 \times 50$ mL TW), the control and JAS1-primed seed lots were established over aqueous solutions in separate p100 Petri-plates with/without dyes at $LD_{50}$ (see the section above) for each of the three dyes. Later, seed germination percentage, shoot, and root length characteristics were recorded in respective tests.

### 2.9. Plant Physiological Growth Assays

Tissue samples from in vitro trials (discussed above) were randomly sampled to carry out various standard biochemical assays viz., total proline (referred in [26]), total chlorophyll and carotenoids [27,28], total phenols and flavonoids [29], and total sugar [30].

### 2.10. Decolorization of Dyed Cellulosic Material

Whatman No.1 cellulosic filter paper (Sigma-Aldrich, New Delhi, India)) and a commercial 100% cellulosic (cotton) textile fabric (purchased from the local textile market in Kharar, Mohali, Punjab) were cut into equally (150 mm × 20 mm) sized strips and dyed using a conventional technique with slight modifications [31]. In brief, the dyed material was dipped at least 10 times individually in 70 °C hot aqueous solution of each azo dye (10.0 g/L) in separate vessels. This was followed by intermittent washing with hot SDW

(70 °C). These materials were left to drying in sunlight for 1 d and underwent overnight oven-drying at 50 °C. Material strips were dipped with one end in 25 mL of MSM inoculated with 4% volume of JAS1 (of an overnight grown inoculum) and with the other end suspended outside the test vessel. Control treatments included dye-supplemented MSM without JAS1 inoculation. Visual decolorization was recorded the next day after removing the material strips from vessels and washing thrice with SDW, followed by drying in sunlight for 4 h.

### 2.11. Statistical and Computational Analysis

All tests were performed at least thrice with three or more replicates per trial. Data values and standard error in graphs, respectively, depict the mean of the values and standard deviations from replicates of three or more trials. FTIR profiles were studied with Essential FTIR® Spectroscopy Software Toolbox (https://www.essentialftir.com/index.html (accessed on 23 November 2022)), and figures were prepared in MS Excel and MS Powerpoint.

## 3. Results

### 3.1. Azo Dye Decolorization by JAS1

Upon inoculating JAS1 in the MSM solutions supplemented with each of the dyes (0.10 g/L of MO, CR, and RB-5) (structures are shown in Figure 1a), the dye color faded with time, resulting in their complete decolorization (as shown in Figure 1b,c), while the control samples did not exhibit any color change. JAS1-assisted decolorization of MO was significantly faster than that of RB-5 and CR (as shown in Figure 1c), where more than 50% MO decolorizing was attained within 48 h of treatment, while it took nearly another 24 h in the case of RB-5 and CR. These results were spectrophotometrically verified, as shown in Figure 1d.

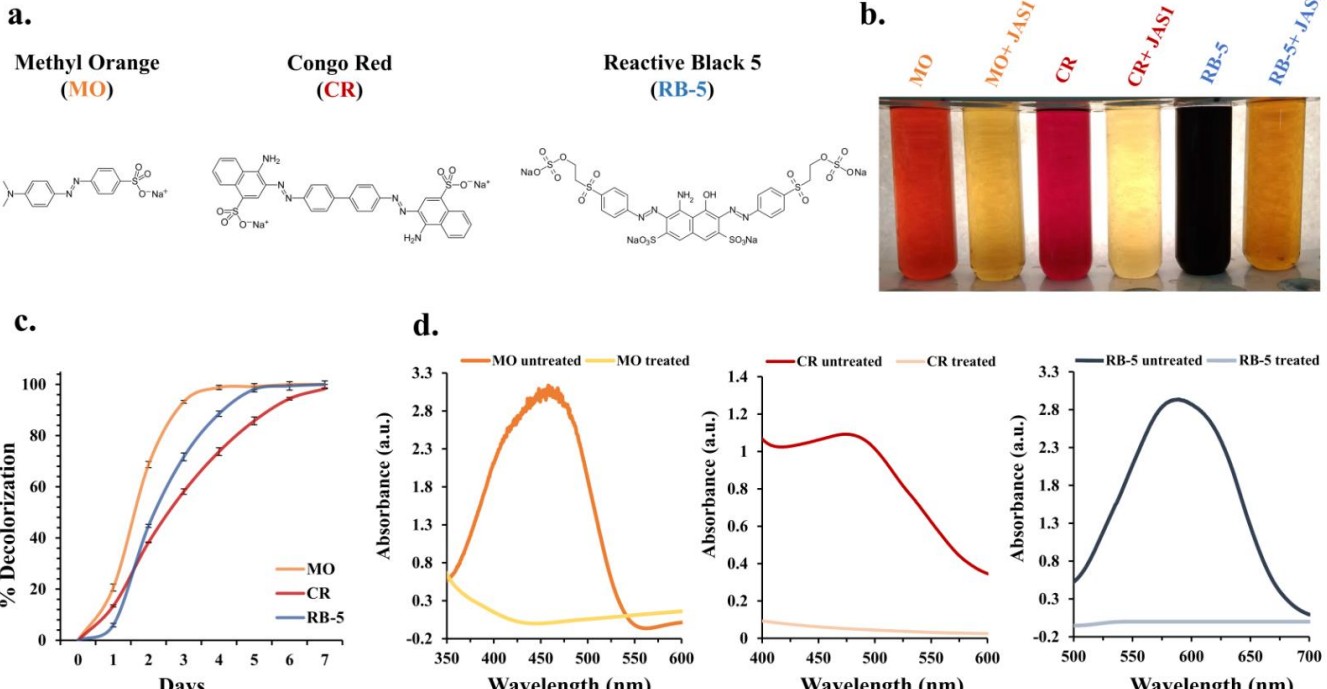

**Figure 1.** In-solution dye decolorization by JAS1. (**a**) chemical structure of the three azo dyes; (**b**) supernatants from each of the in-solution dyes (100 mg/L) treated/untreated with JAS1; (**c**), dye decolorization trends for the three *dyes*; (**d**), spectrophotometry of untreated and treated dyes.

*3.2. FTIR Analyses*

In Figure 2a, the FTIR of the supernatants from control samples with JAS1-untreated MO dye showed characteristic peaks at 693, 620, and 572 cm$^{-1}$ corresponding to C-S stretching vibrations [32]. The sulphonic nature of MO was confirmed with a peak at 1365 cm$^{-1}$. The azo bond (N=N) in MO was confirmed with a peak at 1605 cm$^{-1}$. Aromatic benzene-ring-stretching was observed at 816 cm$^{-1}$. Peaks at 945 and 1036 inferred vibrations. Peaks at 1519 and 1443 cm$^{-1}$ inferred C-H bending at the C=C-H plane. The peak at 2900 cm$^{-1}$ corresponds to CH$_3$ asymmetric stretching, while the peak at 3417 cm$^{-1}$ represents N-H-stretching vibrations. In treated supernatants, the peak at 3468 cm$^{-1}$ shows the N-H-stretching vibrations, and the peak at 2937 cm$^{-1}$ shows asymmetrical CH$_3$ stretching vibrations. The CH$_3$ vibration stretch occurred at 1454 cm$^{-1}$ while CN stretching vibrations were observed at a peak of 1580 cm$^{-1}$. Variations in the FTIR peak profiles in both samples suggested the degradation of MO by JAS1 treatment. These peaks' characterization agreed with previous studies detailing bacteria-assisted MO degradation [32].

In Figure 2b, the FTIR of supernatants from control samples with JAS1-untreated CR dye showed characteristic CR peaks. The peak at 649 cm$^{-1}$ corresponds to the C-C bending vibrations from the aromatic ring; the peak at 1583 cm$^{-1}$ corresponds to the azo (N=N) group in CR. Other than the peak at 832 cm$^{-1}$ for *p*-disubstituted ring vibrations, that at 1064 cm$^{-1}$ corresponds to S=O stretching vibrations for the sulphonic group, and the 1349 cm$^{-1}$ C-N bending vibrations, C=C stretching vibrations from the aromatic ring occurred at 1447 cm$^{-1}$, 3075 cm$^{-1}$ -OH stretching vibrations, and 3469 cm$^{-1}$ N-H stretching vibrations. In treated supernatants, the absence of a peak at 1583 cm$^{-1}$ and stretching vibrations through 645–831 cm$^{-1}$ indicated cleavage of the azo group and the degradation of aromatic amines, respectively. Similar results were reported in studies with other microbes that degrade CR [6,11,33–36].

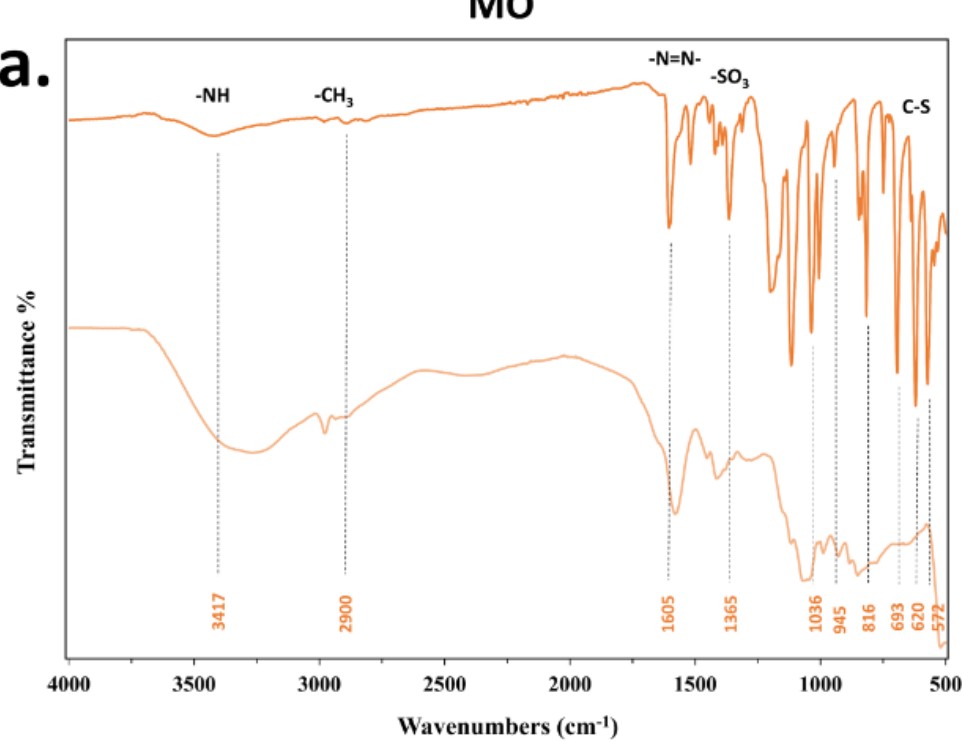

**Figure 2.** *Cont.*

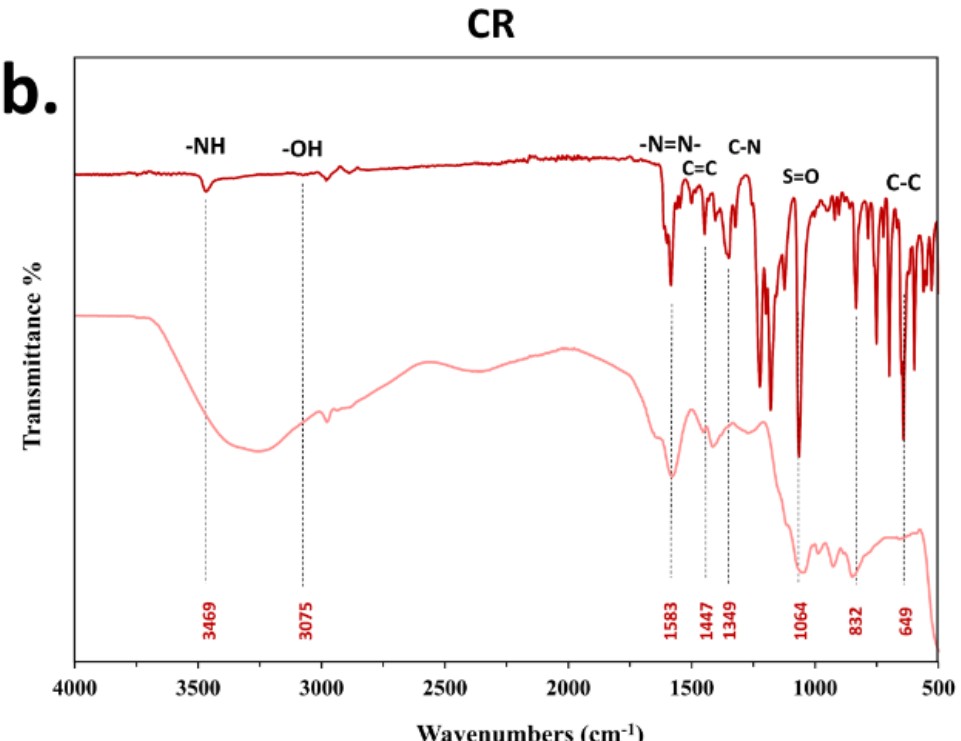

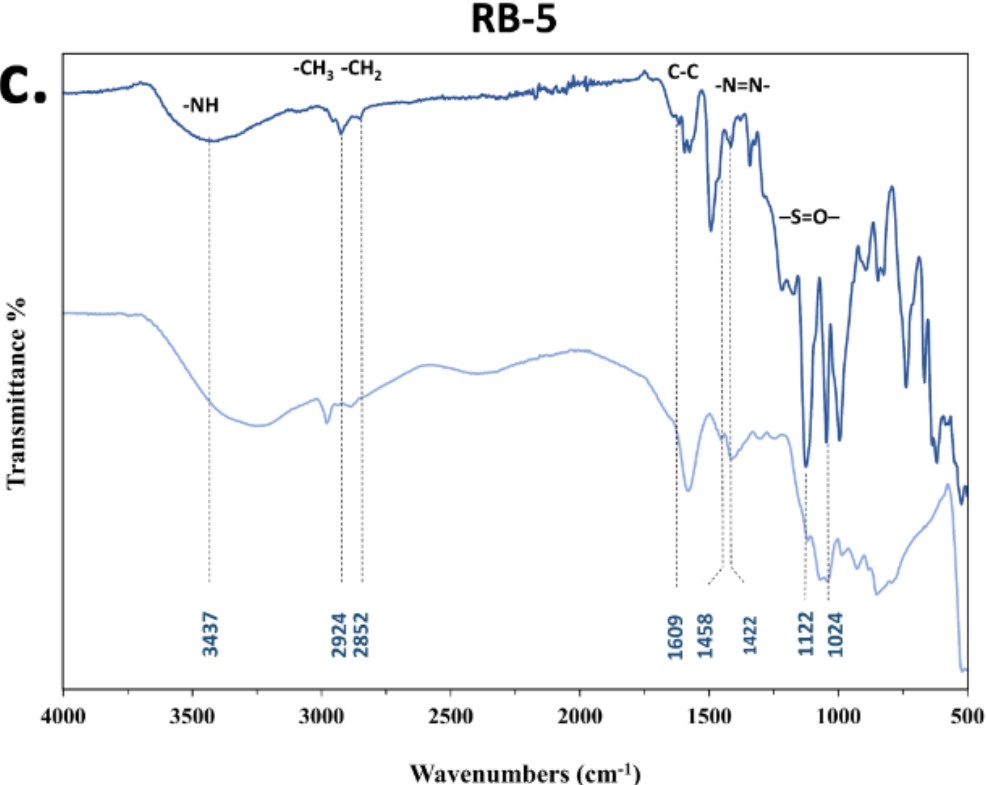

**Figure 2.** FTIR analyses of the three azo dyes treated and untreated with JAS1. (**a**) Supernatant from dye decolorization trials respective to methyl orange (MO); that in (**b**) for Congo red (CR); and in (**c**) for reactive black 5 (RB-5). Dark-colored trend lines correspond to untreated dye and those in faint color correspond to JAS1-decolorized dye sample. Wavenumbers are highlighted for blueprint peaks corresponding to entities in untreated dye.

In Figure 2c, the FTIR of the supernatants from control samples with JAS1-untreated RB-5 dye showed characteristic peaks at 3437 cm$^{-1}$, corresponding to -NH-stretching vibrations respective to amines (expected in the range 3000–3750 cm$^{-1}$) [37]. Peaks at 2924 and 2852 cm$^{-1}$ depict symmetric and asymmetric vibrational stretching from -CH$_3$ and asymmetric stretching vibrations in -CH$_2$. Next, a peak at 1609 cm$^{-1}$ characterized the C-C vibration from the aromatic ring, while that at 1458 cm$^{-1}$ evidenced the presence of azo (-N=N-) linkages [38]. The vibrational stretch for -S=O- corresponds to the peak at 1123 cm$^{-1}$ which, along with the peak at 1024 cm$^{-1}$, confirms the sulfoxide group in the untreated RB-5 sample. Supernatants from test samples with JAS1-decolorized RB-5 detected differences in spectra compared to that in the control sample (Figure 2a–c). The absence of many characteristic peaks and the presence of newer peaks possibly resulted from the RB-5 degradation products. For example, peaks ranging from 1750 to 2800 cm$^{-1}$ (peaks at 2660, 2521, 2441, 2318, 2280, and 2171 cm$^{-1}$) were quite eminent in the untreated samples, which could not be detected in JAS1-treated samples. There was a reduction in peak corresponding to -NH stretching vibrations, and an apparent absence of the peak from wavenumber range 3437 cm$^{-1}$ (in JAS1-untreated sample) and occurrence of a new peak at 3232 cm$^{-1}$ (in JAS1-treated sample) which indicates the reduction in the azo group (-N=N-). At the low-frequency region 620–850 cm$^{-1}$, the intensity of the peaks significantly decreased, inferring the break-up of aromatic rings [39]. These spectral data suggest significant the structural deconstruction of RB-5 during its JAS1-led decolorization. Similar results were reported in many other RB-5 decolorization studies [37,38,40–43].

### 3.3. TLC Verified Dye Decolorization

TLC confirmed the complete decolorization of each dye. Chromatographs for each of the JAS1-treated and -untreated dye solutions were compared with the water-dissolved solution of the original dye (Figure 3). The chromatographic front of JAS1-treated dye sample lanes (respective to each of the three dyes) detected no spot. Observation under UV-light inferred trails, possibly from dye degradation. Other studies reported similar TLC profiles [44].

### 3.4. Optimization of Dye Decolorization by JAS1

In our study, the decolorization efficacy of JAS1 was studied in the effect of various C-source and N-sources in separate tests. Changes in other variables such as pH, incubation temperature, inoculum size, initial dye concentration, and agitator speed were also investigated. The results are shown in Figure 4. Glucose was found to be critical for JAS1, showing maximum decolorization of the three dyes (MO, CR, and RB-5) within 96 h and almost half within 48 h of incubation. In previous studies, we observed an influence of media compositions on the colony morphology, growth, and motility of JAS1 [21]. Among all nitrogen sources (N-sources), potassium nitrate enhanced MO's decolorization rate, while peptone favored CR and RB-5. At culture incubations of 28 °C (an optimum incubation temperature for most species in the genus *Agrobacterium*), JAS1 also showed the highest kinetic rates for dye decolorization. A pH below 7.0 affected the overall dye decolorization rate. Azo dyes are known to produce toxicity to microbes above a specific concentration. In our study, we found that JAS1 could tolerate very high initial dye concentrations, which affected their decolorization rate (Figure 5). An increase in the orbital shaking speed (from 50 to 150 rpm) significantly improved the overall dye decolorization efficiency, showing the critical role of the mechanical effect. A higher inoculum percentage favored higher rates of dye decolorization, suggesting the possibility of increased in-solution activity(s) of dye-degrading enzymes of JAS1. Considering the optimized parameters, we achieved 80–90% decolorization of the three dyes within 24 h (Figure 4).

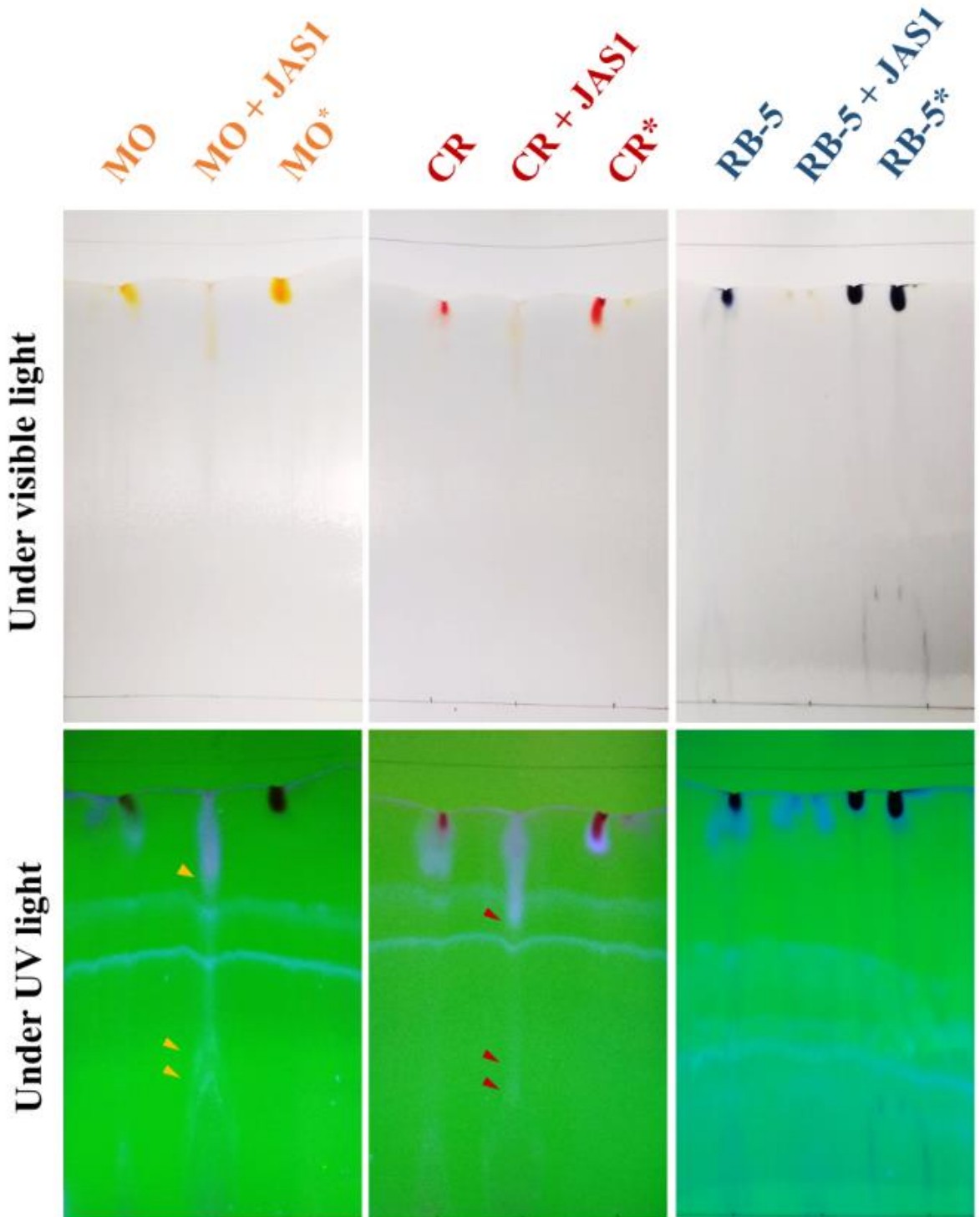

**Figure 3.** TLC analyses of the extracted supernatants from trials with the three dyes. Lanes developed from spots made with processed supernatants hailing from dye decolorization trials in the absence and presence of JAS1 in-solution respective to each of the three dyes: MO, CR, and RB-5. * Superscripts-labeled lanes indicate water-solubilized original dye, used as a reference standard. The chromatograph front showed under both visible and UV light. Possible dye degradation products are shown with colored arrows where marked.

## a. MO

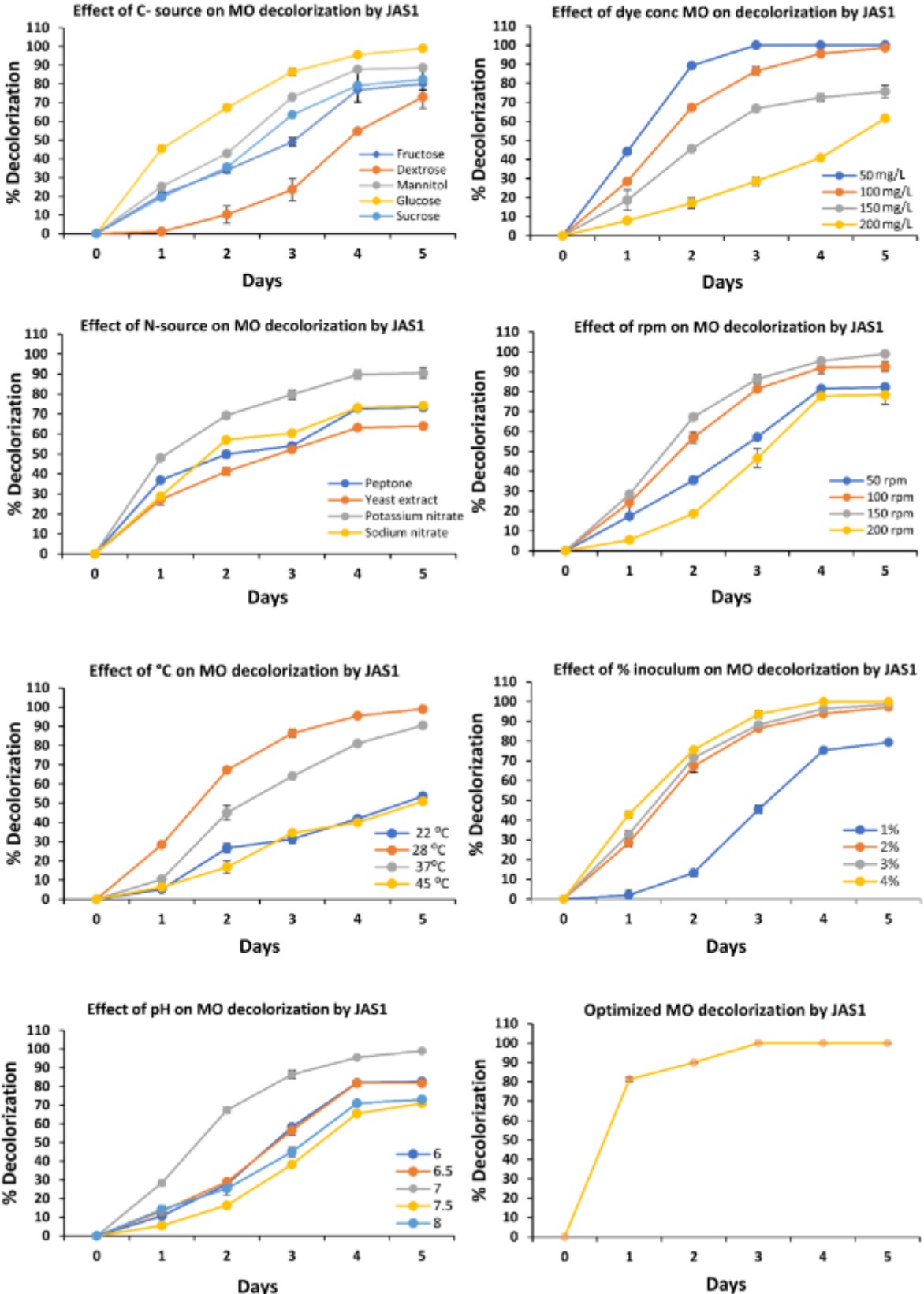

**Figure 4.** *Cont*.

## b. CR

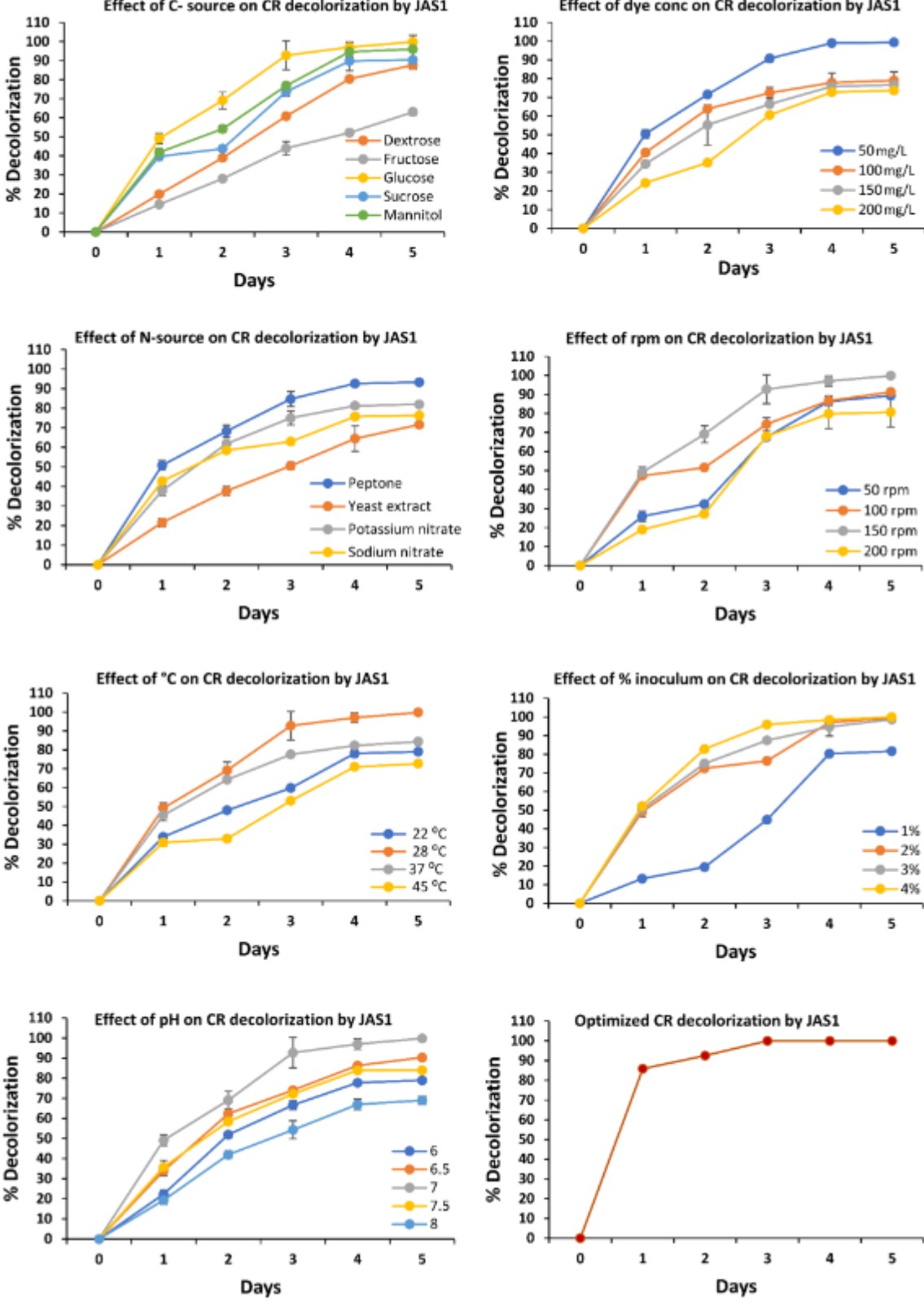

**Figure 4.** *Cont.*

## c. RB-5

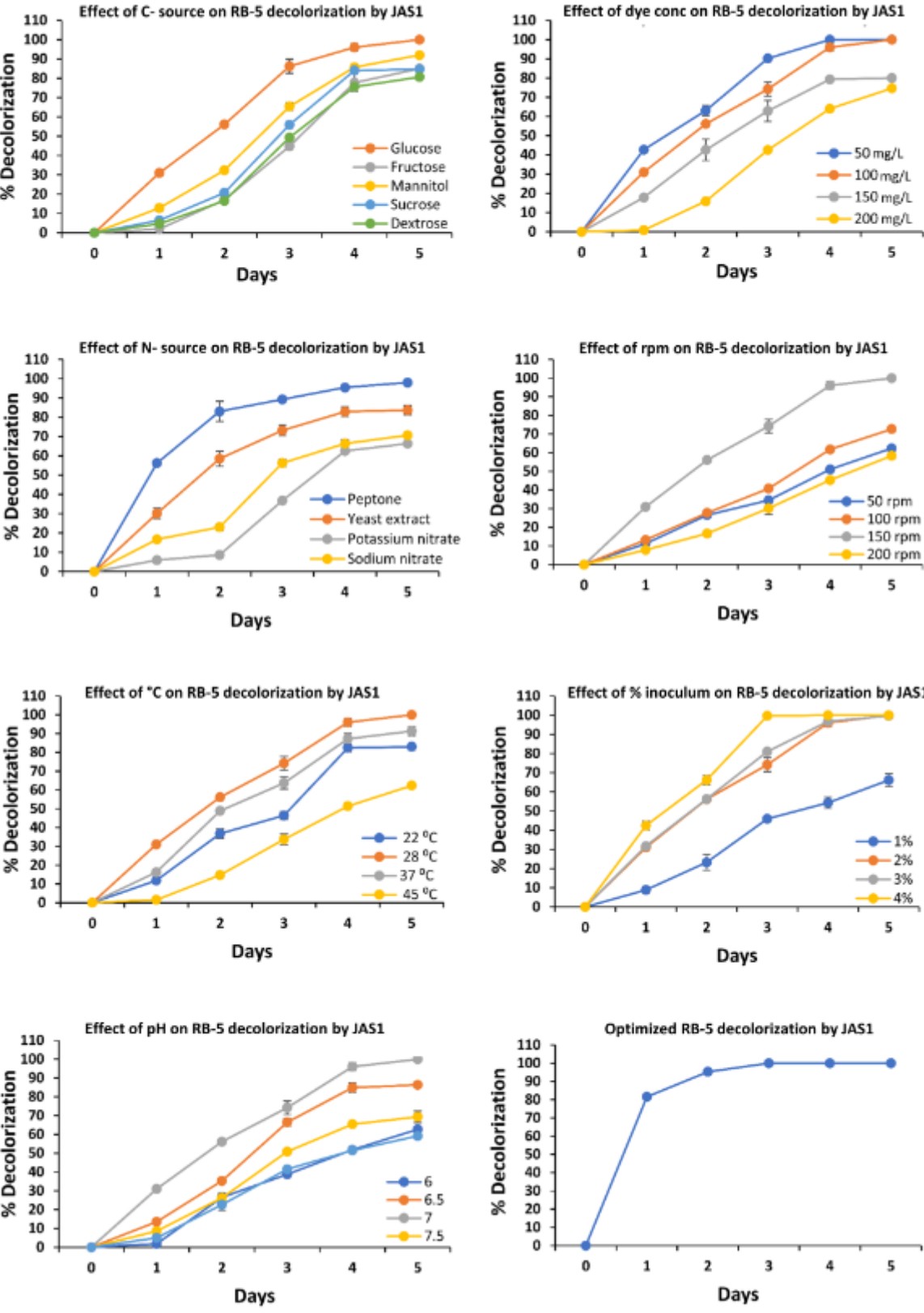

**Figure 4.** Physicochemical optimization of in-solution azo dye decolorization efficiency of JAS1. MSM solutions carrying either of the three dyes (100 mg/L of MO (**a**); CR (**b**); RB-5 (**c**)) and inoculated with

$1 \times 10^6$ CFUs/mL of the overnight-grown fresh culture of JAS1 were separately treated with variations in aforesaid physical and chemical parameters. All tests were repeated thrice with three replicates per test in one trial. Computed decolorization percentage values and their standard deviations depict means of the means emanating from the three trials. See Materials and Methods for more details.

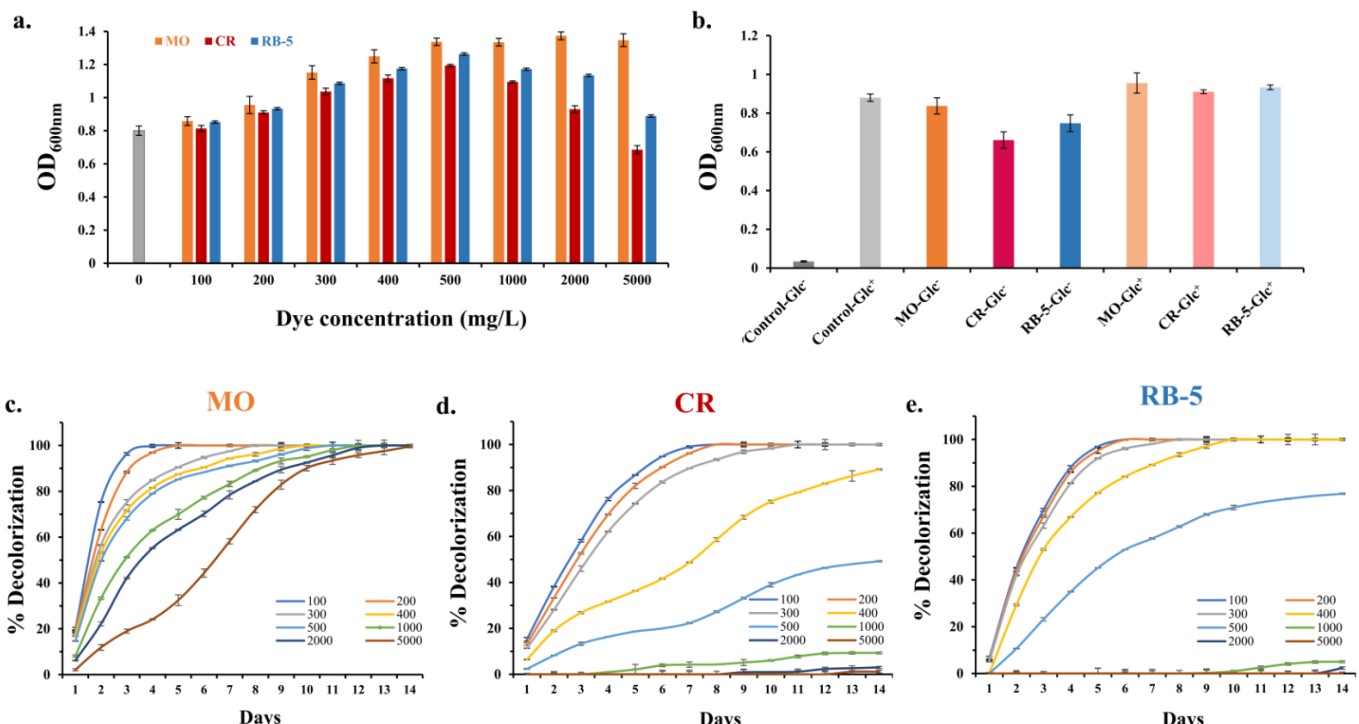

**Figure 5.** JAS1 tolerance and decolorization extent for the three azo dyes. (**a**) Growth of JAS1 inoculum ($1 \times 10^6$ CFU mL$^{-1}$) at increasing in-solution concentrations (100–5000 mg/L) of three azo dyes, MO, CR, and RB-5, in separate tests; (**b**) the effect of C-source (glucose, 0.6%) supplementation (+gluc) and its removal (-gluc) on the growth of JAS1 inoculum ($1 \times 10^6$ CFU mL$^{-1}$) in a solution containing various dyes (200 mg/L) in separate tests; (**c–e**) decolorization efficiency of JAS1 over two weeks for a range of initial concentrations (100–5000 mg/L) for each of the three azo dyes: MO, CR, and RB-5, respectively.

### 3.5. JAS1 Can Tolerate and Decolorize High Azo Dye Concentration

In our observation, the growth of JAS1 was found in wide concentrations (up to 5.0 g/L) for each dye (as shown in Figure 5a). JAS1 tolerated MO concentrations of up to 5.5 g/L and showed effective decolorization until 5.0 g/L within two weeks of incubation (as Figure 5c). After that, JAS1 showed no growth above 5.5 g/L of MO. Interestingly, JAS1 decolorized the CR and RB-5 solutions up to 0.4 g/L, and was unable to decolorize at any higher concentration, even when left for a prolonged time (Figure 5d,e). However, experimental results (as discussed in Figure 5a) on culture density increments showed that JAS1 could tolerate CR and RB-5 levels of up to 5.0 g/L, indicating a sustained cellular mechanism critical that is for its survival even in harsh chemical environments, although there is not enough genetic expression to produce those enzymes which assist in azo dye degradation. In support of such inferences, we observed that the viability of JAS1 was affected in those experiments of CR (>0.5 g/L) and RB-5 (>0.45 g/L) even when high culture densities were recorded. The distinctive biodegradability of JAS1 towards these azo dyes (MO, CR, and RB-5) could be subjective based on the structural differences between these dyes. Other factors, such as degradation products that were formed over the time in-solution, are compete for specific enzymes and limit the growth

of JAS1 (bacteriostatic effect) (Supplementary Figure S1), while bactericidal effects were observed at higher concentrations MO (>5.5 g/L), CR (>0.5 g/L) and RB-5 (>4.5 g/L).

### 3.6. JAS1 Can Use Azo Dyes as a Sole Carbon Source for Growth but Critically Requires a Co-Substrate for Dye Decolorization

Based on our observation (as shown in Figure 5a), an unusual growth (higher cell density) of JAS1 in the presence of dyes compared to the control sample was recorded. This cellular growth of JAS1 suggested that bacteria were able to consume glucose + dye as a carbon source compared to the control sample, which only consumed glucose. This inference is rare, as typical bacteria are incapable of utilizing azo dyes as a sole carbon source due to the electron-withdrawing feature of azo dyes [45], and only a few are known to consume azo dyes [10,38], while others consume the degradation products of dyes (mainly amine derivatives, phenols, and carboxylate sulphonates) as a carbon and nitrogen source [46]. In decolorization assays, MSM contained 0.6% glucose as a standard carbon source and 0.1% ammonium chloride as a nitrogen source, where JAS1 growth kinetics were found to be progressive. To our surprise, JAS1 showed growth in a glucose-deficient solution without any dye decolorization, indicating that its growth is dye-concentration dependent (as shown in Figure 5b). Additionally, solutions with glucose showed dye decolorization, suggesting that the biodegradation activity of JAS1 requires glucose as a co-substrate. However, how dyes exclusively support the growth of JAS1 in the absence of glucose remains unclear.

### 3.7. Azo Dye Decolorization by JAS1 Is Probably Enzyme-Catalyzed

In another experiment, we studied whether dye decolorization is affected by some size-fractionated entity(s) (Figure 6a). the cell-free supernatant of the JAS1 culture was dialyzed using a cellulose acetate membrane (MWCO 14 kDa). The retentates and permeates were mixed with 0.05 g/L of dyes (Figure 6a,b). As expected, only the retentates showed a complete decolorization of three dyes and not the permeates (as shown in Figure 6b,c). Interestingly, the dye decolorization rate appeared to improve with increasing retentate strength (as shown in Figure 6d), suggesting that the dye decolorization effectuates exclusively due to certain high-molecular-weight entities (>14 Kda) in the retentate and not the permeate. Additionally, dye-degrading enzymes such as lignin peroxidase, manganese peroxidase, laccase, veratryl alcohol oxidase, tyrosinases, aminopyrine N-demethylase, DCIP reductase, and azoreductases are high-molecular-weight (>14 Kda) entities [46–54]. The kinetic rate of MO decolorization was found to be faster than that of RB-5 and CR (Figure 6c), also in agreement with our results, as shown in Figure 5.

### 3.8. Congo Red Adsorbs to the JAS1 Surface

Following the complete *in-solution* decolorization of the three dyes using JAS1 treatment, the bacterial pellet showed dye-specific red coloration only in the case of CR (Supplementary Figure S1). This suggests that the decolorization of CR by JAS1 effectuates an adsorption phenomenon in addition to its enzymatic degradation. Many studies have previously reported CR decolorization by microbes following its surface adsorption [7,36,55–57]. Moreover, many studies report the specific use of CR to identify *Agrobacterium* isolates that produce curdlan (an EPS) because of its affinity with these sugars [58,59]. To reiterate, JAS1 is an *A. pusense* strain that also produces EPS [21].

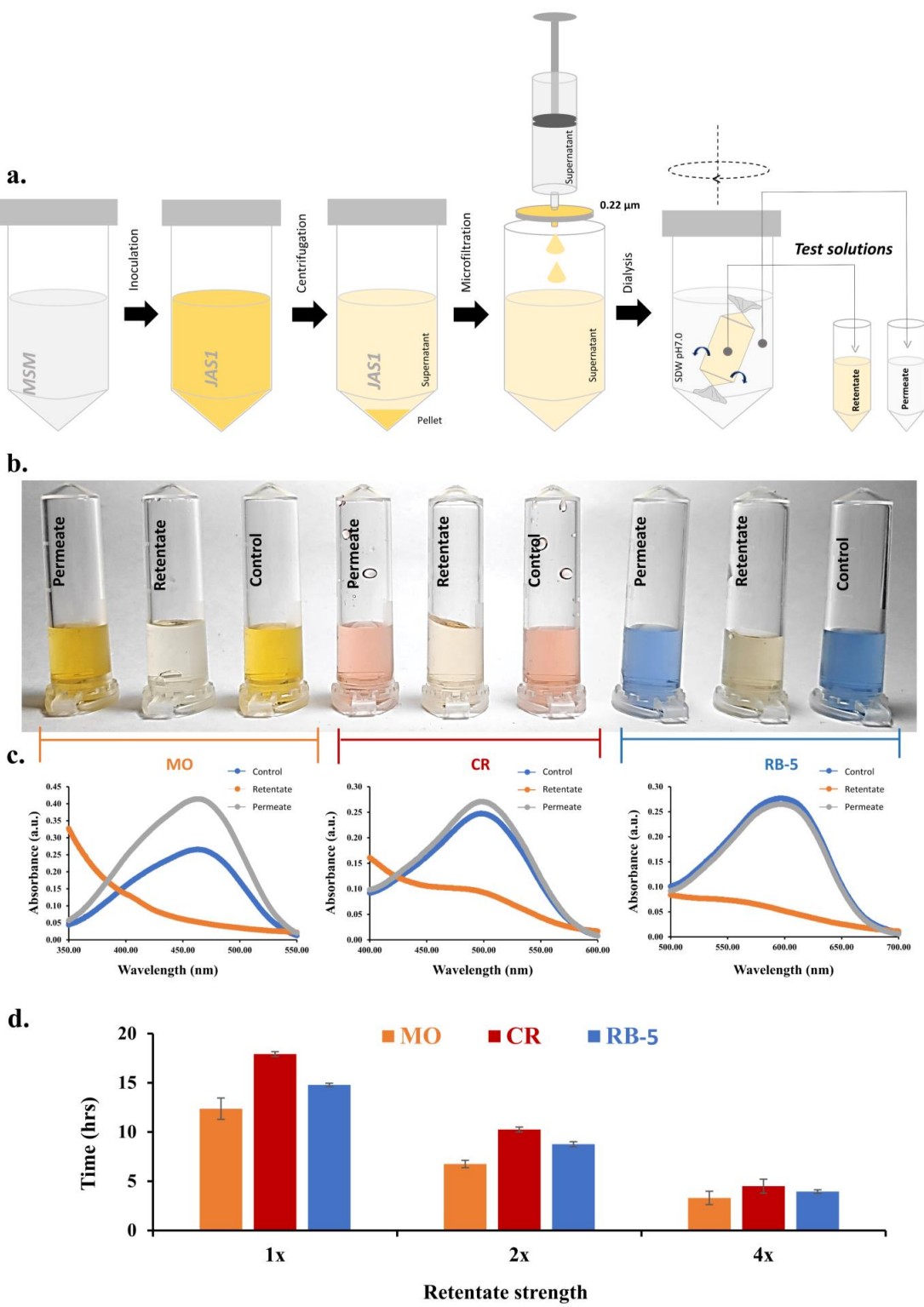

**Figure 6.** The probable enzymatic mode of azo dye decolorization by JAS1. (**a**) Cell-free supernatants of JAS1 prepared via centrifugation and subsequent ultrafiltration were cleared using dialysis tubing (pore size 12–14 kDa MWCO) for 4 days, after which permeates and retentate solutions of the tubing were tested for dye decolorization; (**b**) apparent decolorization of the three dyes only in the retentate of the tubing (as shown in (**a**)); (**c**) absorbance reads of retentate, permeate, and control solutions (as shown in panels (**a**,**b**) post 24 h incubation at RT and controls at wavelength maxima for each of the dyes; (**d**) increasing strength of JAS1-free supernatant expediates complete decolorization rate of each of the three dyes.

### 3.9. Probable Azo-Dye Dynamics under in solution Treatments with JAS1

The above results showed that JAS1 showed variability in its ability to tolerate and decolorize three structurally distinctive azo dyes (MO, CR, RB-5). The results suggest various generalizations regarding the potential dynamics of dyes and their degradation by JAS1 (Figure 7). At a low inoculum size, the bacterial cell surface was occupied with CR, only allowing for cell division to occur. However, with a high concentration of CR/RB-5 in the solution, surfaces in both the parent cells and daughter cells were occupied with these dyes. Higher concentrations of dyes in the solution can lower their rate of decolorization. This can be attributed to the unsuitability of the surface-to-dye ratio, the toxicity of degradation products from dyes, and/or the inhibition of active sites in azo-reducing enzymes [60–65]. The limited lack of decolorization (CR and RB-5) could possibly be due to the direct or indirect (by-product-led) repression and/or inhibition of the expression and/or secretion of enzymes responsible for degradation-governed decolorization. Higher concentrations of azo dye substrates can block enzyme binding sites and reduce the degradation rate [66]. Alternatively, dye reduction rates may also proportionately depend on bacterial growth rate [67] and microbial toxicity [68]. However, in the case of MO, the simplest form of the azo dye structure is probably easily accessible to enzymes secreted by JAS1 *in solution*. At lower concentrations (0.05–0.20 g/L) of dye decolorization and/or its optimization assays, we do not observe the repressive/inhibitory influence from CR and RB-5. This could be due to their more complex structures (Figures 1b, 4 and 5). However, the dye decolorization efficacy can be seen to improve with increases in inoculum size. After adding fresh and increased JAS1 inoculum and/or its cell-free supernatant, the dye decolorization is readily affected. This proves the initially observed attenuation/repressive effect of the dyes. Even though RB-5 is more complex in structure, it is readily decolorized *in solution*, similar to MO. This is probably due to the action of enzymes and occurs in the presence of a co-substrate. The dynamics of CR, however, are apparently more complex. CR is known to self-associate by the parallel stacking of aromatic rings to form ribbon-like supramolecular assemblies [69,70]. Other than these, it intercalates within proteins by sandwiching amongst peptide chains, forming large oligomers up to approx. 28 m [71,72]. Under the reducing conditions of SDS PAGE, free Congo Red has been shown to run as a 50 KDa-sized entity [72]. Binding to proteins involves both hydrophobic and electrostatic interactions [72]. CR is believed to structurally mimic co-enzymes in binding to various proteins and enzymes such as RNA-polymerase, elastin, dehydrogenases, and kinases [73–76]. Other than this, in yeast, CR treatment is known to affect both cell wall biogenesis and cell morphogenesis by forming chains of connected cells with staggered walls [77]. CR is known to inhibit chitin biosynthesis in fungi [78,79]. In relation, however, to its effect on bacteria, only a few studies vouch that CR can bind cell walls and/or agrobacteria-secreted exopolysaccharides [59,80]. CR binding has also been often renounced in assays distinguishing *Agrobacterium* isolates from those in the genus *Rhizobium* [81–83]. We reiterate that our JAS1 isolate is an exopolysaccharide-producing *A. pusense* strain [21]; hence, its decolorization activity over CR via surface adsorption can is considerable. The same enzymatic degradation can occur with any *Agrobacterium* species, as shown in our experiments (Figure 6). With the complex behaviors of CR, its internalization into the bacterial cell may be considered unfavorable.

The above experiments show that JAS1 can tolerate high levels of the tested azo dyes (MO, CR, RB-5) and use them as a preferred carbon source for growth. However, high concentrations of the complex azo dyes CR and RB-5 negatively affect the dye decolorization efficacy of JAS1. The initial glucose concentration might have caused the initial growth of JAS1 and, after depletion, the dyes were consumed as an alternative carbon source.

### 3.10. Azo Dyes Dose-Dependently Impose Toxicity over Seed Germination and Plant Growth

To study the impact of in-solution decolorized water (as a result of JAS1 treatment) on sustaining life forms, we ascertained dose(s) respective to the three azo dye(s) that critically affect wheat seed germination and seedling growth profile in seven days. The

seed germination percentage reduced significantly with the increasing concentration of each azo dye in the germination media. CR was found to be more toxic, completely ceasing seed germination at 0.60 g/L, while a 1.0 g/L concentration was found to be critical to seeds in the case of both MO and RB-5 (Figure 8a). Seedling growth at a subcritical dye dosage also witnessed negativities regarding the reduced length of the shoot (Figure 8b) and root (Figure 8c). Root surfaces were stained with dye color. This intensified at increasing dye doses, but at subcritical (for germination) dye concentrations, roots appeared stiff with patchy darkening. Studies in the literature also confirm that textile dyes have such detrimental effects on plant growth [11,84–90].

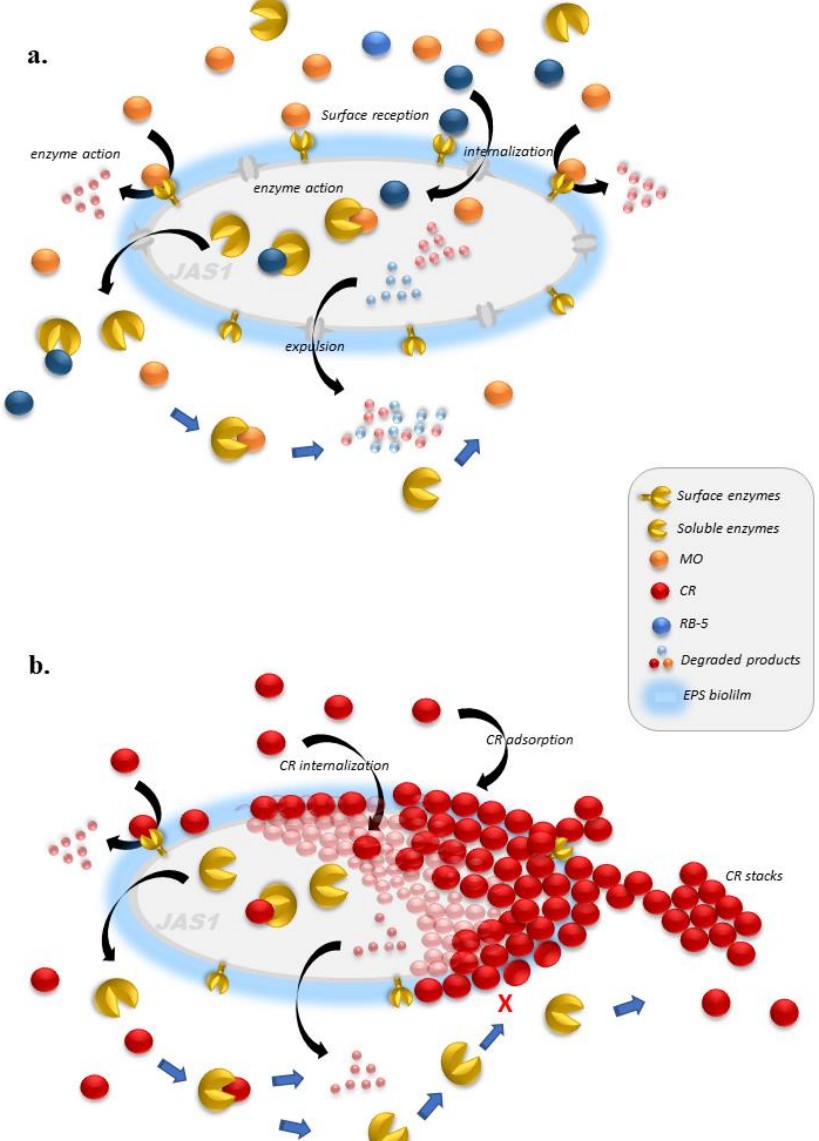

**Figure 7.** Probable dye dynamics in-solution and dye footfall over JAS1 influencing dye degradation and/or adsorptive decolorization. The figure shows the hypothetical process of azo dye degradation by secretory and surface processes from JAS1, influenced by the dye, degradation by-products, and dye-degrading enzymes. The easy dye decolorization of MO and RB-5 is shown on the upper panel (**a**) with dynamics around a JAS1 cell in solution. These dyes may be easily processed by cells, both intracellularly and extracellularly, by the action of putative enzymes that cause MO and RB-5 degradation. In the case of CR, shown in the lower panel (**b**) with another JAS1 cell, dye affinities to cell surface entities such as proteins and sugars on the cell wall and EPS, and the CR's self-stacking features, drastically complicate the possible JAS1 decolorization mechanisms. Such complications

may deter CR entry into cells, as also seen with membrane caging experiments in this manuscript as well as CR remanence over plant roots (see texts describing this here and in relevant upcoming sections).

### 3.11. By-Products from JAS1-Degraded Azo Dyes Are Non-Toxic

To assess the possible adverse effects of JAS1-degraded dye products, we studied their effect on wheat seed germination and seedlings. From the previously elucidated effects from the three azo dyes on in vitro seed germination and seedling growth profiles (Figure 8), we chose respective threshold doses near $LD_{100}$ of MO (0.8 g/L), CR (0.5 g/L), and RB-5 (0.8 g/L) that accounted for a nearly completely abolished seed germination. Supernatants were collected after the JAS1-assisted complete decolorization of each of these dye doses in separate tests and were used as seed germination media. As expected, after a week's observation, the untreated dyes severely affected the seed viability, resulting in a poor germination response (Figure 9a,b). Many studies have shown similar detrimental effects from various dyes on plant seed germination and seedling growth [11,85–91]. Seeds in the dye-decolorized (JAS1-treated) supernatants germinated well without any negativities (Figure 9a,b). These results suggested that JAS1 pretreatment could rescue wheat seeds from any possible detrimental effect of the non-biodegradable products of azo dyes. These effects equate with responses seen in supernatants from the control tests (Figures 8 and 9b). Additionally, physiological growth assays over the seedlings validate these outcomes (Figure 9c). This plant-rescuing feature of JAS1 probably follows from the complete dye degradation, for example, the neutralization or conversion of any toxic by-products, such as aromatic amines, to biologically stable entities. The outcomes of this experiment (Figure 9) and FTIR analyses (Figure 2) support the JAS1-assisted complete degradation of the three azo dyes and/or their metabolization into consumable sources (C-source and N-source).

### 3.12. JAS1 Priming Rescues Plants from Dye Adversities and Maintains Overall Health

We evaluated the performance of seed-priming treatments with JAS1 over the productive growth of wheat in-solution with the azo dyes. To do this, we selected a nearby $LD_{50}$ specific to each of the three azo dyes (MO, 0.60 g/L; CR, 0.30 g/L; and RB-5, 0.50 g/L) (Figure 8). Over a week-long trial, as expected, the JAS1-primed seeds showed improvements in seed germination and seedling growth. Seed germination showed significant increments in shoot and root lengths compared to the control tests with JAS1-unprimed seed lots (Figure 10a,c). In Figure 10b, root sections from different treatments show the dye-decolorization effect from JAS1. In unprimed seedlings, each of the dyes strongly adheres to the root surfaces, which, besides their toxic effects, reduces the water uptake. However, in contrast to the JAS1-unprimed seed lots, seedling roots emanating from JAS1-primed seeds could easily recover from these detriments. This can be surmised from a comparison of the physiological growth analyses in seedlings (Figure 10c).

### 3.13. JAS1 Can Remove Azo Dyes from Different Cellulosic Substrates

To study the effectivity of this approach, we considered using cellulose materials, such as dyed paper and textile strips with each of the three dyes and then treated them with JAS1 by dipping one end in the media with the other end suspended outside the test vessel (as presented in Figure 11). As expected, both the MO- and RB-5-dyed textile and paper strips resulted in an apparently complete decolorization within one day. CR decolorization on dyed textile strips, however, could not be witnessed at all, and that on paper strips was only sparingly visible even after a week-long incubation (Figure 11). We showed that JAS1 treatment accounts for the in-solution adsorption of CR, in addition to its supposed enzymatic degradation (Figure 6).

### 3.14. Cellulosic Membrane-Caging of JAS1 and/or Its Spent Supernatant to Treat Azo Dyes

The effluent sludge from dye and textile industries, however, contains complex mixtures of chemicals other than dyes. However, in an open environment, such water pollution may allow for the growth of a mixed microorganism population [92]. The complex sub-

stances in the effluent from dye and textile industries may negatively affect the effectiveness and viability of JAS1 and other dye-degradation microbes. Therefore, in our interest, we attempted to translate JAS1's ability to decolorize dyes using a cellulosic membrane caging/tubing filled with JAS1. We conducted separate tests for each azo dye (MO, CR, and RB-5) to see if JAS1 and/or its cell-free spent supernatant held in dialysis tubing would effectively decolorize the dyes. Our caging system is composed of a cellulose-based membrane (MWCO 12–14 KDa). This membrane would selectively permeate various molecular entities such as dyes, their degraded products, and other azo-dye-reducing equivalents. It would, however, not allow for the escape of the putative dye-degrading enzymes. We used 0.40 g/L of each of the dyes, given that, at this dosage, JAS1 could show fair viability (Figure 5a) and steady decolorization of the three azo dyes (Figure 5c–e).

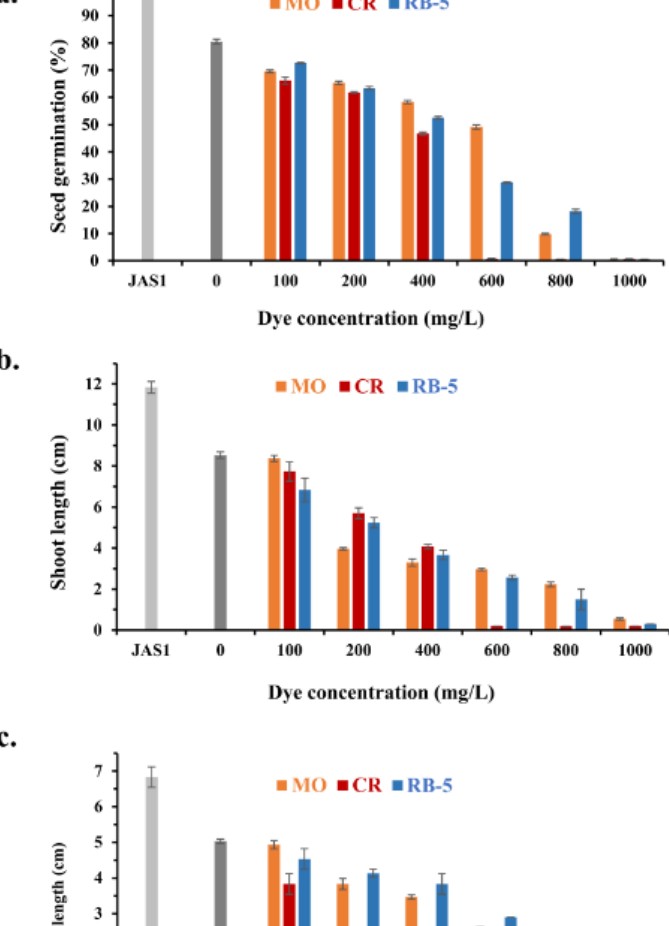

**Figure 8.** Effect of azo dyes on wheat seed germination and plant growth profile. (**a**) Results from separate tests of wheat seed germination under treatment with a range of the three azo dye concentrations (MO, CR, and RB-5); (**b**,**c**) The concomitant effect on the shoot and root length as growth parameters. Observations were recorded 7 days after treatments. Treatment alone with JAS1 (shown with gray columns) shows a relatively enhanced plant growth profile as compared to dye-free control treatment (shown with dark gray column).

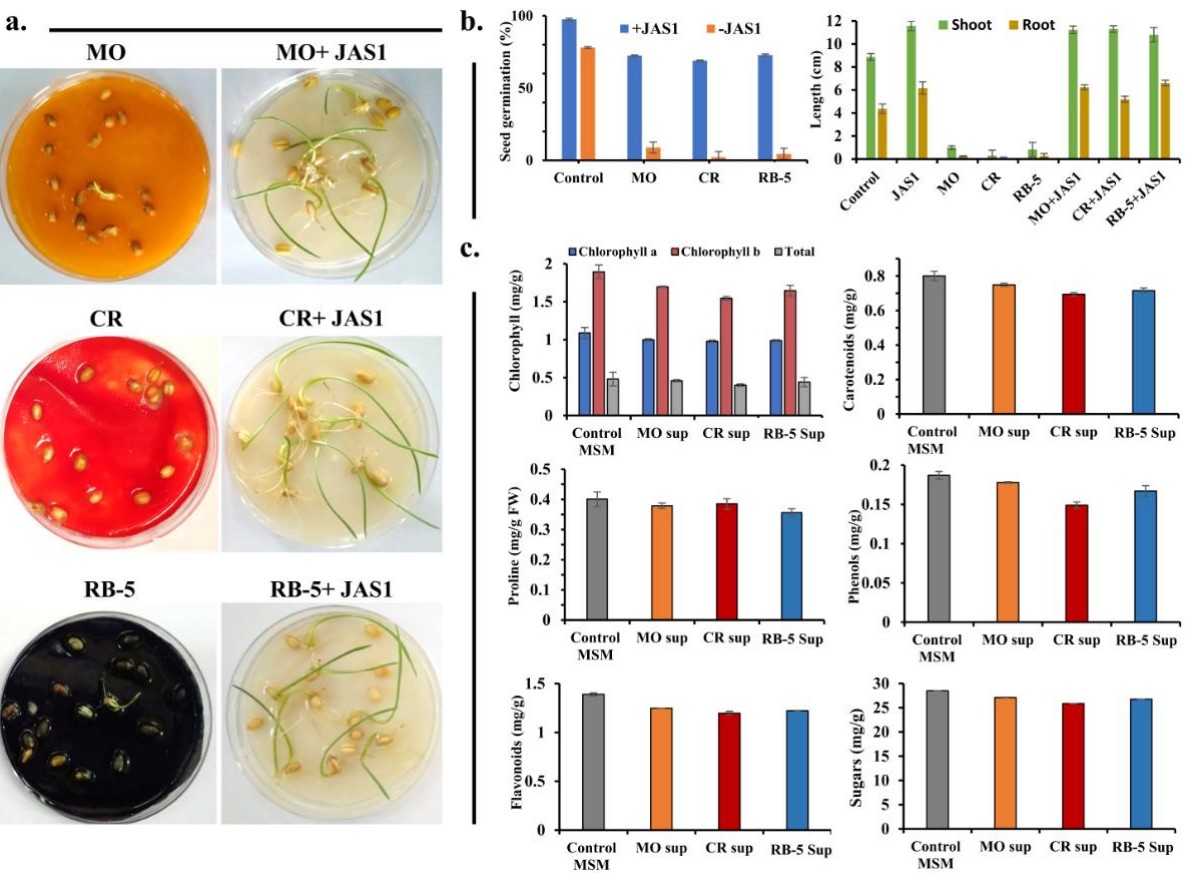

**Figure 9.** Effect of by-products from JAS1-assisted dye decolorization on wheat seeds. (**a**) Effect of nearby $LD_{100}$ on each of the three azo dyes and that of their JAS-degraded products (cell-free supernatants) compared to wheat seeds; (**b**) seed germination and plant growth metrics for seedlings (as in (**a**)); and (**c**) physiological growth measurements for various seedlings with their growth under different tests (as in (**a**,**b**)). See the preceding text for details.

As expected, dipping treatments with cellulose caging showed significant decolorization of all dyes in separate tests (Figure 12). MO was completely decolorized by both the cellulose-caged JAS1, and its cell-free supernatant compared to control tests after about five days of incubation (Figure 12). Neither the caging nor the pellet retained any MO-like coloration that would otherwise suggest adsorption (Figure 12a) at any instance within or beyond the trial periods. This, therefore, suggests that MO could readily and wholesomely pass through into the caging for action by JAS1 and/or its supernatant. Another possibility could be the externalization of small, putative, dye-reducing equivalents (from the pellet and supernatant caging), which could effectuate the in-solution action over the dyes. This, however, was crosschecked in the previous experiment and was not the case (Section 3.7; Figure 6). RB-5 was also seen to be completely decolorized in-solution by effects in the caged pellet and also those in the caged supernatant (as shown in Figure 12). Similar to the case shown above with MO, neither the pellet nor the cellulose cage showed any RB-5 coloration. In the case of CR, as expected, a low dye decolorization (Figure 12) could be recorded in-solution, possibly indicating saturation under prompt adsorption over the cellulosic cage. Any further decolorization beyond the trial period could not be witnessed in either cage (Pellet and Sup. cages), justifying the probable clogging of pores under high CR binding and CR's self-stacking, as discussed before [55,69,71,72,77,93].

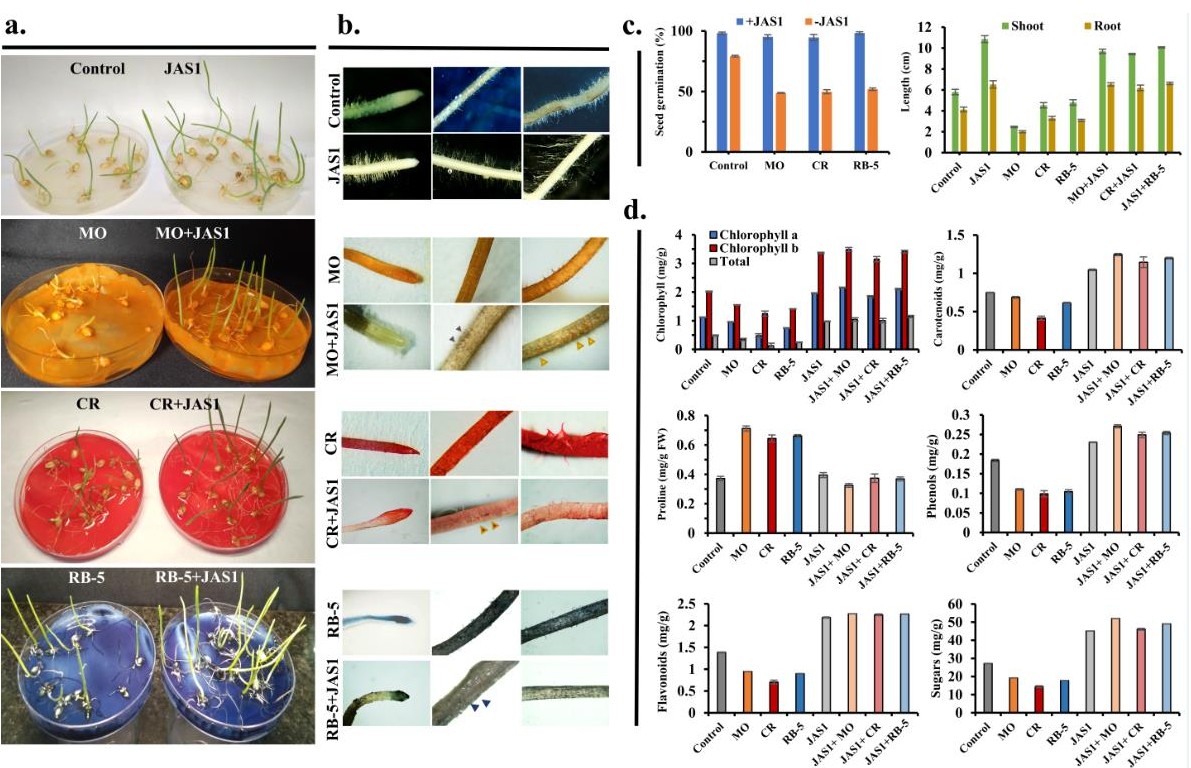

**Figure 10.** Effect of JAS1 priming on wheat seed germination in dyes. (**a**) JAS-primed and/or -unprimed wheat seeds were shown to germinate in aqueous solutions with nearby LD$_{50}$ dosages of either or none of the three dyes; (**b**) the effect of JAS1 on roots in seedlings following seed treatments (shown as in (**a**)); (**c**) seed germination and plant growth metrics; and (**d**) physiological growth measurements for various seedlings along with their growth under different tests (as in (**a**,**c**)). See the preceding text for details.

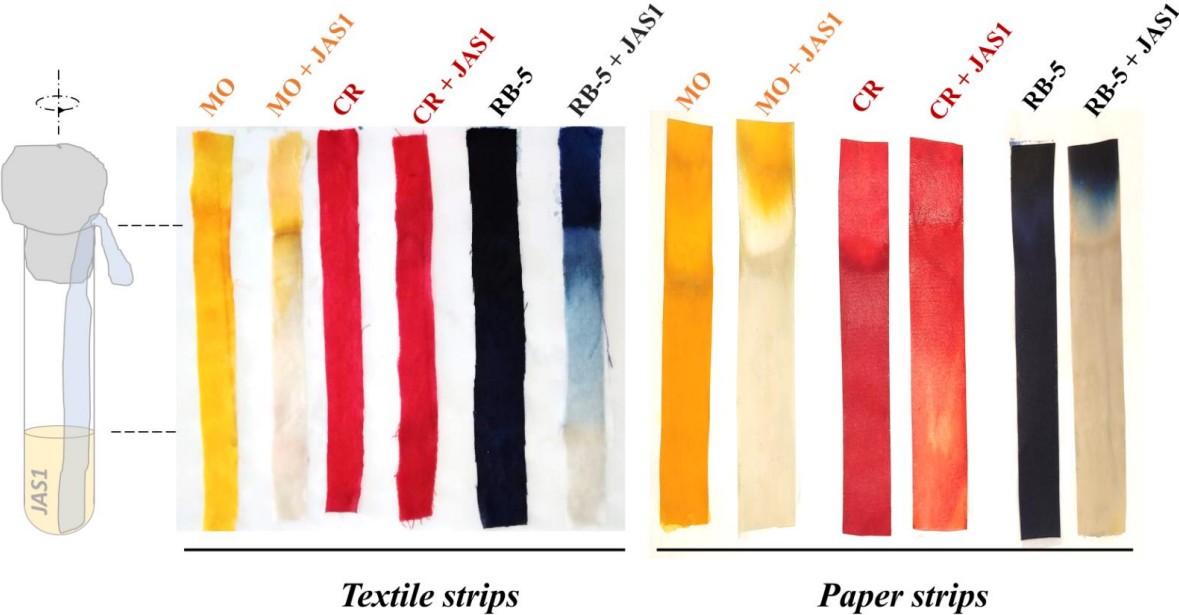

**Figure 11.** Dye decolorization by JAS1 on textile and paper strips. Separate textile and paper strips dyed with either MO, CR, or RB-5 were dipped in MSM with and/without JAS1 inoculum. Note the sparingly visible decolorization on CR-dyed paper under the effect of JAS1; otherwise, negligible decolorization occurred on the CR-dyed textile strip.

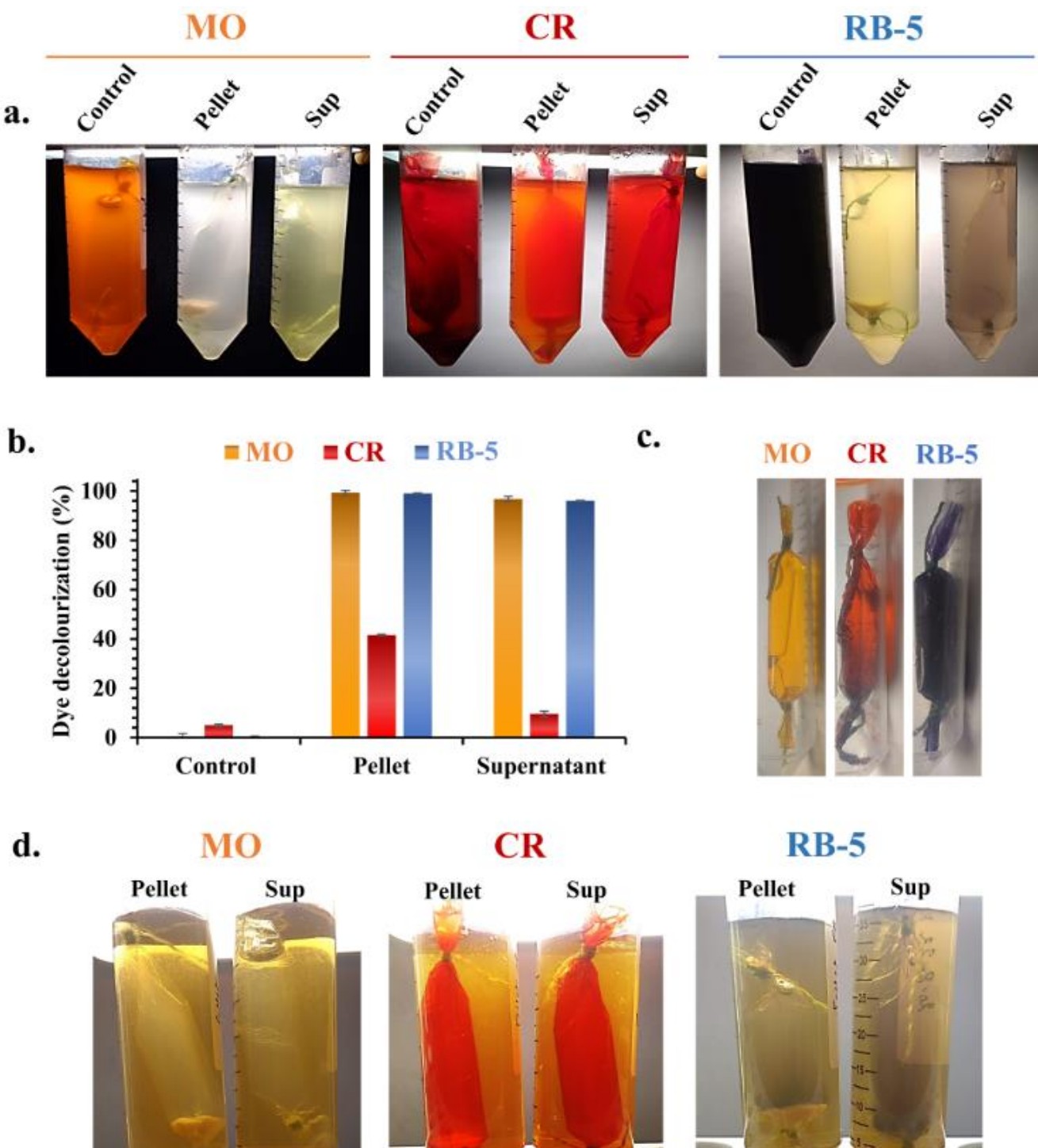

**Figure 12.** Effect of cellulosic membrane caging functionalized with JAS1 pellet and/or its cell-free supernatant on dye decolorization. (**a**) Observations at the 5th-day post-incubation, where JAS1 pellet (labeled Pellet) and/or supernatant (labeled Sup) was held in dialysis caging (MWCO 12–14 KDa) and dipped in-solution with each of the dyes (MO, CR, and RB-5) in separate tests; (**b**) dye decolorization extent (from tests as shown in (**a**)). Note the limited decolorization of CR in respective Pellet and Sup caging. Note also that control cages were packed only with pure MSM, processed similarly to supernatants in the Sup cages. (**c**) Control cages were removed and rinsed thrice in SDW but still showed binding dyes on the surface. Although not shown here, it is worth noting that a visual check

for solutions in these controls' cages justified the easy internalization of both MO and RB-5; however, that for CR was very limited, and, upon dipping these cages in JAS1-inoculated MSM, only MO and RB-5 could readily be decolorized, not CR (as in the effect seen with CR-dyed textile fiber and paper; see Figure 11). In (**d**), cages as shown in **a** were removed in the post-trial period and visualized as dipped in MSM to later prepare for the reproducibility of the dye treatment. Note that CR is bound to the membrane and is not easily treated by JAS1 as before.

## 4. Conclusions

Bioprospecting studies, specifically with microbial PGP isolates from pollution-resistant succulent plants, might offer potential dye-bioremediating agents. Although bacterial species belonging to many genera were reported to enzymatically degrade and decolorize textile dyes, only one study in the genus *Agrobacterium*, that of *A. radiobacter*, was documented as having attribute [12]. This is in addition the canonical absorption feature of some EPS-binding dyes, such as CR, by almost all *Agrobacterium* species, which was prevalently only used for their taxonomic identification. *Agrobacterium pusense* strains, however, have not been tested for such applications to date. Our studies with JAS1 provide new information on the *A. pusense* strain's ability to decolorize structurally different azo dyes (Methyl Orange, Congo Red, and Reactive Black 5). The in-solution dye-degrading potential of JAS1 would allow for eco-friendly, and cost- and energy-efficient effluent treatment. Other than this, its dye-decolorization efficacy on various cellulosic substrates indicates the potential use of the JAS1 as a de-inking and decolorizing agent in the recycling and reusing of processed paper and textile products, respectively. As a PGP endophyte, as previously investigated by us, JAS1 can colocalize plant roots and deliver enhanced seedling growth, which can improve the water-retention capacity of the soil [21]. The dye decolorization and its facilitation of the seed-germination attributes of JAS1 would thus prove useful for prospective agricultural crop cultivation by farmers, who will inevitably use dye-contaminated irrigation water from the textile and related industries in the near vicinity.

**Supplementary Materials:** The following supporting information can be downloaded at: https://www.mdpi.com/article/10.3390/w15091664/s1, Figure S1: JAS1 pellet post in-solution dye degradation. In (a), except CR which retained its color on the bacterial pellet, MO and RB-5 could not show any colored pellet (following from Figure 1b in the manuscript). (b), JAS1 inoculum from these pellets could easily regrow as streaks over NA.

**Author Contributions:** G.M. and J.K. conceptualized the studies, and together performed all the tests; J.T. and S.S., helped with recordkeeping and resources during experiments; G.B.S., J.C.B.K. and S.D. validated the studies, discussed the results, and helped with outsourcing a few analyses; M.A.W., M.F.K., helped with resources; J.K., G.M., J.R., A.N. and K.K.K. formulated the first draft of the manuscript; G.M., J.K. and A.N. compiled all data, and together wrote the final version of the manuscript; G.M. supervised all the experiments. All authors have read and agreed to the published version of the manuscript.

**Funding:** The authors express their sincere appreciation to the Researchers Supporting Project Number (RSP2023R466) King Saud University, Riyadh, Saudi Arabia.

**Data Availability Statement:** The bacterial isolate JAS1 used in the study was a previously characterized novel strain of *Agrobacterium pusense* isolated from leaf tissue cultures of a succulent plant *Sansevieria trifasciata* var. *falcata* L. (Prain), as reported by us [21,22] with its deduced 16SrRNA, atpD, and recA gene sequence homologs deposited in NCBI's GenBank (accession number MW827601, MZ741443 and MZ741444 respectively). Any raw data supporting the conclusion of this article will be made available by the authors, without undue reservation, and upon formal request.

**Acknowledgments:** The authors express their sincere appreciation to the Researchers Supporting Project Number (RSP2023R466) King Saud University, Riyadh, Saudi Arabia. The authors also thank the UIBT, Chandigarh University for infrastructural support and for registering JK as a doctoral student under the able supervision of GM. The institute, however, had no role in shaping the manuscript and/or in the decision to publish.

**Conflicts of Interest:** The authors declare no conflict of interest. The funders had no role in the design of the study; in the collection, analyses, or interpretation of data; in the writing of the manuscript; or in the decision to publish the results.

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
