# Peer review of "Reactive Black-5, Congo Red and Methyl Orange: Chemical Degradation of Azo-Dyes by Agrobacterium"

_water, doi:10.3390/w15091664_

Round 1

Reviewer 1 Report

In this paper, Kaur et al performed comprehensive work on a novel application of Agrobacterium pusense strain JAS1 on bioremediating toxic azo dye pollutants inevitably used by the global textile industries. The authors have very well described the decolorization attributes in-solution and remarkably have also shown the dye remediation feasibility by using both the bacterial culture and cell-free spent culture supernatant. They also tested the putative dye-degradation by-products as being non-toxic and stable to farm-grown commercial plants. Additionally, the JAS1 strain being a plant growth-promoting bacteria showed its remarkable application in priming with seeds reproducing the above effects in context to productively up-leveling productivity features of the crop and dually as well negating the irrigatable clean water crisis at farms close to dying giants. The biofunctionalized cellulose casing system has as well been shown for use in effluent treatment plants in dyeing relevant sites.  

The manuscript is well written, there are some minor issues that need to be addressed.

-As the authors cited many studies which highlighted pollution remediation using Sansevieria plants, did they attempt to test whether sansevieria plants in their hand demonstrated any decolorization of these dyes?

-Does the decolorization on dyed textiles and paper strips also effected in treatment with crude purification of cell-free supernatant of JAS1? This is probable though with that shown in the case of decolorization on dyed cellulosic cases.

-For border visibility of the work, the authors recommended to highlight the role of the United Nations SDG goal for the current manuscript.

-The paper has some typographical errors and should be revised accordingly.

-Line, 23; “reactive back 5” should be corrected to “reactive black 5”

Author Response

Reviewer 1 Comments and Suggestions for Authors (response to reviewer’s comments ahead in blue text)

In this paper, Kaur et al performed comprehensive work on a novel application of Agrobacterium pusense strain JAS1 on bioremediating toxic azo dye pollutants inevitably used by the global textile industries. The authors have very well described the decolorization attributes in-solution and remarkably have also shown the dye remediation feasibility by using both the bacterial culture and cell-free spent culture supernatant. They also tested the putative dye-degradation by-products as being non-toxic and stable to farm-grown commercial plants. Additionally, the JAS1 strain being a plant growth-promoting bacteria showed its remarkable application in priming with seeds reproducing the above effects in context to productively up-leveling productivity features of the crop and dually as well negating the irrigatable clean water crisis at farms close to dying giants. The biofunctionalized cellulose casing system has as well been shown for use in effluent treatment plants in dyeing relevant sites.  

The manuscript is well written, there are some minor issues that need to be addressed.

Repones to reviewer’s comments:

The authors of the manuscript thank the reviewer for constructive comments, and thought-provoking queries which we address herewith point by point (in blue text):

  1. As the authors cited many studies which highlighted pollution remediation using Sansevieria plants, did they attempt to test whether sansevieria plants in their hand demonstrated any decolorization of these dyes?
  2. Yes, we do find dye decolorization attributes in Sansevieria plants as well. Coincidently, trials with many Sansevieria trifasciata varieties are ongoing in this regard at our end in conjunction with azo dye degradation and we are attempting as well to exploit Sansevieria cellulosic biomass at co-bioprospecting with JAS1 and other endophytes isolated from these varieties to provide a comparative evaluation of efficacious azo dye degradation and to design more able, eco-friendly, and sustainable management of dye waste. However, we would be presenting these in our future manuscript taking into view the focus of this manuscript exclusively on the potential of JAS1 at dually benefitting crop plants with dye remediation and PGP properties.
  3. Does the decolorization on dyed textiles and paper strips also effected in treatment with crude purification of cell-free supernatant of JAS1? This is probable though with that shown in the case of decolorization on dyed cellulosic cases.
  4. Yes, dye decolorization on paper and textile fabric was equally efficient under treatment with cell-free spent supernatant of JAS1. Yes, this has been also exemplified in trials with cellulosic caging of JAS1 and its supernatant in the manuscript. We intend to extend this knowledge to test the decolorization of various other dyes and commercially dyed paper and fabric substrates using this bacterial isolate.
  5. For border visibility of the work, the authors recommended to highlight the role of the United Nations SDG goal for the current manuscript.
  6. The authors thank the reviewer for this suggestion. We incorporated relevant UN-SDGs that the outcomes of the work in this revised manuscript would potentially suffice.
  7. The paper has some typographical errors and should be revised accordingly.
  8. We have corrected the typographical errors and revised the manuscript as suggested.
  9. Line, 23; “reactive back 5” should be corrected to “reactive black 5”
  10. We have incorporated the correction in the revised version of the manuscript.

Reviewer 2 Report

The manuscript "Efficient in-solution and functionalized cellulose-caging assisted azo dye degradation using a novel Agrobacterium pusense strain" requires some proper editing (English and overall structure) before considering for publication. 

some comments,

1.      Abstract is not clear and too generalized. Please add numeric data.

2.      Please correct sentence structure and spelling mistakes, such as affintive (line 19) docuemt (line 24). This comment applies to all parts of the manuscript.

3.      Introduction part should be in short and meaningful paragraphs. Split the first two paragraphs.

4.      Introduction lacks UpToDate work on dye removal via cellulosic materials. Refer to https://doi.org/10.1016/j.jhazmat.2021.126958. Please revise the introduction.

5.      Please put heading levels and numbering throughout the manuscript for better understanding.

6.      Figure legends, scale, scale, text, etc. are not uniform. Please revise.

7.   Figures 2, 4, 5, 6, and 8: It is not possible to read and follow. Please improve the resolution.

8.      There is too much text in figure 13. Please revise.

Author Response

REVIEWER 2 Comments and Suggestions for Authors

The manuscript "Efficient in-solution and functionalized cellulose-caging assisted azo dye degradation using a novel Agrobacterium pusense strain" requires some proper editing (English and overall structure) before considering for publication. 

We thank the reviewer for critical evaluations and constructive comments. The English language and structure have been edited and improvised in the revised version of the manuscript.

some comments,

  1. Abstract is not clear and too generalized. Please add numeric data.

We have completely changed the abstract as per the above suggestions.

  1. Please correct sentence structure and spelling mistakes, such as affintive (line 19) docuemt (line 24). This comment applies to all parts of the manuscript.

We have incorporated these and other corrections in the revised version of the manuscript.

  1. Introduction part should be in short and meaningful paragraphs. Split the first two paragraphs.

The introduction section has been edited according to the above suggestions.

  1. Introduction lacks UpToDate work on dye removal via cellulosic materials. Refer to https://doi.org/10.1016/j.jhazmat.2021.126958. Please revise the introduction.

We have incorporated the relevant works on dye removal via cellulosic materials and cited the above reference as well.

  1. Please put heading levels and numbering throughout the manuscript for better understanding.

These edits can be now seen in the revised manuscript.

  1. Figure legends, scale, scale, text, etc. are not uniform. Please revise.

-We corrected these in the revised manuscript.

  1. Figures 2, 4, 5, 6, and 8: It is not possible to read and follow. Please improve the resolution.

Figure files have been separately enclosed on the submission portal. We presume that the reviewer must have raised this issue with the PDF version of the manuscript (we apologize for the inconvenience), the word version secured the high resolutions however. We have now maintained the high resolution of the figures in the hereby submitted revised PDF file of the manuscript.   The pdf-converter erroneously compressed the images. We request still to tally with the MS-word file or the images if accessible in the submission portal.

  1. There is too much text in figure 13. Please revise.

We decided to remove this figure from the manuscript, considering the other outcomes in the follow-up of this work, and would present it in the future manuscript.
